# Molecular dynamics study of differential effects of serotonin-2A-receptor (5-HT$_{2A}$R) modulators

**Jordy Peeters**[1], **Dimitri De Bundel**[2], **Kenno Vanommeslaeghe**[1]*

**1** Department of Analytical Chemistry, Applied Chemometrics and Molecular Modelling (FABI), Vrije Universiteit Brussel (VUB), Brussels, Belgium, **2** Experimental Pharmacology (EFAR) group, Vrije Universiteit Brussel (VUB), Brussels, Belgium

* public@kenno.org

## Abstract

The serotonin-2A receptor (5-HT$_{2A}$R) is an interesting target for drug design in the context of antidepressants that might have a rapid onset of action and/or be effective in treatment-resistant cases. The main challenge, however, is that the activation of this receptor can provoke hallucinations. Recent studies have shown that activating the receptor with certain (partial) agonists could potentially give rise to antidepressant effect without hallucinogenic side effects. Although substantial research has been done in this area, the atomistic details of this differential activation of the serotonin-2A receptor are not fully understood. In the present study we performed multiple atomistic molecular dynamics (MD) simulations on 5-HT$_{2A}$R bound with two antipsychotics, three different potential non-hallucinogens and two hallucinogens to identify the receptor's ligand dependent conformations. Overall, our findings suggest that modest 5-HT$_{2A}$R activation would only yield antidepressant effects and hallucinations result from excessive activation. While modest activation through microdosing may be problematic on account of abuse potential as well as possibly narrow and patient-dependent therapeutic windows, modest activation through administration of a sufficiently weak partial agonist may offer a viable drug development pathway.

## Author summary

Activating the serotonin-2A receptor (5-HT$_{2A}$R) with specific agonists is a promising strategy for the development of a new class of antidepressants. However, potent agonists such as LSD, DMT or psylocibin generally cause hallucinations. To aid the development of 5-HT$_{2A}$R-targeted antidepressants that do not cause this side effect, we seek to gain a better understanding of the receptor by studying its activation mechanism. Specifically, we performed molecular dynamics simulations to explore how different drugs interact with 5-HT$_{2A}$R, both in the absence and presence of an intracellular binding partner. This led to the identification of

**Data availability statement:** The timeseries and trajectories are made available on Zenodo at http://doi.org/10.5281/zenodo.15031692.

**Funding:** This work was supported by the Vrije Universiteit Brussel (VUB, https://www.vub.be/en ) through research council (OZR) starting funds (OZR2893, awarded to KV). The super-computing resources and services used in this work were provided by the VUB and the VSC (Flemish Supercomputer Center, https://www.vscentrum.be/ ), the latter being funded by the FWO and the Flemish Government, through project 2023_100 (awarded to KV). The funders did not have any role in the study design, data collection and analysis, decision to publish or preparation of the manuscript.

**Competing interests:** The authors have declared that no competing interests exist.

tentative intermediate states along the activation pathway, which could be linked to the ligands' pharmacological properties. Our findings suggest that hallucinogens cause an excessive build-up of activated receptors, whereas carefully designed mild activators could lead to a new generation of antidepressants that do not induce hallucinations.

## 1. Introduction

The 5-HT$_{2A}$-receptor belongs to the family of G protein-coupled receptors (GPCRs) and is a crucial component of the serotoninergic system, implicated in various mental health conditions including schizophrenia, bipolar disorders, ADHD, migraine, and depression-related disorders. [1–3] In treating schizophrenia and bipolar disorders, the 5-HT$_{2A}$R is primarily targeted with atypical antipsychotics acting as either antagonist [4–10] or inverse agonist. [4,7,8,11–13] Conversely, the use of 5-HT$_{2A}$ agonists is limited due to their propensity for inducing hallucinogenic effects. Since the 1950s, depressive disorders have been treated with tricyclic antidepressants (TCAs), mono-amine oxidase inhibitors (MAOIs), selective serotonin reuptake inhibitors (SSRIs), and serotonin-norepinephrine reuptake inhibitors (SNRIs), all of which increase the concentration of endogenous serotonin in the synaptic cleft. [14] While clinical studies on hallucinogens remain limited, research on 5-HT$_{2A}$ receptor agonists such as LSD, psilocybin, and mescaline has demonstrated rapid and long-lasting effects in the treatment of depression, substance use disorders, and anxiety. [15–17]. This suggests that direct pharmacological activation of the 5-HT$_{2A}$-receptor could be helpful in the treatment of depression. Accordingly, several non-hallucinogenic 5-HT$_{2A}$ agonists [18,19], including compounds IHCH-7086 [20] and (R)-69, [21] have been reported to exert significant antidepressant but not hallucinogenic activity in mouse models.

The classic GPCR activation mechanism involves the formation of a multi-protein complex that includes the G-protein at the intracellular side of the activated receptor. In this environment, the G-protein is phosphorylated, setting in motion an intracellular signaling cascade. Eventually, the receptor is desensitized, which is thought to involve the blocking of the G-protein binding site by β-arrestin. However, β-arrestins have been shown to also exert influence on downstream signaling pathways, particularly those involving kinases such as the Src/AKT cascade. [22]

It is not fully understood why some 5-HT$_{2A}$ agonists have antidepressant and/or hallucinogenic effects. One of two (not mutually exclusive) hypotheses links these different effects to differential activation of other receptors. [23,24] In this context, Seks-saoui and colleagues showed that the antidepressant-like effects of DOI and lisuride were abolished in constitutive 5-HT$_{2A}$ receptor knockout mice, while those of psilocybin persisted. [25] Notably, these psilocybin effects were not blocked by 5-HT$_{1A}$ or D1/D2 antagonists, suggesting involvement of additional or compensatory pathways.

Moliner and colleagues demonstrated that psilocin and LSD can directly bind to the TrkB receptor, a neurotrophin receptor involved in synaptic plasticity. [26] They further showed that LSD's antidepressant-like effect was not prevented by the 5-HT$_{2A}$

antagonist M100907, implicating TrkB in 5-HT$_{2A}$-independent mechanisms. Conversely, no significant effects of LSD and psilocin on TrkB were observed in a recent study by Jain and colleagues on the subject of the polypharmacology of psychedelics. [27] Also, region-specific targeting studies provide compelling evidence for a critical role of cortical 5-HT$_{2A}$ receptors in psilocybin's action. Shao and colleagues reported that conditional knockout of 5-HT$_{2A}$ receptors in pyramidal neurons of the medial prefrontal cortex eliminated psilocybin's antidepressant-like effects in mice. [28] Jain and colleagues further showed that 41 classical psychedelics activate nearly all other serotonin receptor subtypes as well as dopamine and adrenergic receptors. [27] Since 5-HT$_{2A}$'s role in the action of psychedelics is well established, we interpret the activation of other receptors as evidence that the hallucinogenic potential is further modulated by these other receptors. For example, 5-HT$_{1A}$ has been associated with mediating distinct psychoactive or therapeutic effects. [15,29] Specifically, 5-MeO-DMT and its analogues, which interact strongly with 5-HT$_{1A}$, were shown to exhibit antidepressant-like properties without hallucinogenic effects. Orthogonal to the involvement of other receptors, there exists a substantial body of work studying how different 5-HT$_{2A}$ ligands give rise to different extents of G protein and β-arrestin associated signaling, and how this is linked to differential (antidepressant and/or hallucinogenic) effects. Specifically, Cao and colleagues synthe-sized a potential antidepressant non-hallucinogenic compound IHCH-7086. [20] This ligand exhibits high binding affinity for 5-HT$_{2A}$, does not activate the Gq pathway, and shows low maximal efficacy in their β-arrestin2 recruitment assay. However, the role of β-arrestin in signaling is still a matter of debate. For example, Kaplan and colleagues's partial agonist (R)-69 is biased toward G protein pathway activation ($E_{max} = 87\%$ relative to 5-HT) at mid-nanomolar concentrations, while β-arrestin2 recruitment was detectable only at much higher concentrations. Nevertheless, (R)-69 has also been reported to produce non-hallucinogenic antidepressant effects in mouse models. [21] In another study, Wallach and colleagues demonstrated that an agonist's hallucinogenic potential correlates with its efficacy for the Gq pathway. Specifically, they found that efficacious ligands with an $E_{max}$ below 70% (relative to 5-HT) did not induce the head-twitch response in mice, [30] a known predictor of hallucinogenic activity in humans. [31] Taking these studies together seems to suggest that partial agonists with low efficacies can give rise to antidepressant effects without psychedelic properties regardless of their activation pathway. The present work seeks to contribute to the understanding of these differential effects through a detailed MD study of the receptor itself. Specifically, we aim to determine whether the binding of psychedelic compounds causes the receptor to populate different conformational states compared to non-psychedelic ligands. To this end, we mapped the conformations that become accessible to the receptor in response to extracellular binding of different ligands and intracellular binding of a construct that mimics the Gq protein. In addition, we performed a number of simulations where we perturbed the experimental starting structure (by removal of the G protein or by docking a different ligand) and observed conformational transitions. Trends in these observations yielded new insights into the microscopic functioning of this multifaceted receptor as well as corroborating hypotheses that have been proposed based on experimental evidence.

## Molecular activation mechanism

The general activation mechanism of GPCRs has been studied extensively. On a macroscopic scale, an outward move-ment of Transmembrane Helix 6 (TM6) occurs upon activation, accompanied with an inward movement of TM3 and TM7 as shown in Fig 1. Note that the Ballesteros-Weinstein nomenclature is employed in further sections. [32] At the level of individual interactions between side chains (Fig 1A), the activation of GPCRs involves breaking the ionic lock of the E/DRY motif between residues Glu$^{6x30}$ of TM6 and Arg$^{3x50}$ of TM3. Additionally, the hydrophobic interaction between Ile$^{3x46}$ and Leu$^{6x37}$ in these helices also breaks. This allows TM7 to rearrange itself into a kinked helix so its NPxxY motif (Asn$^{7x49}$, Pro$^{7x50}$ and Tyr$^{7x53}$) interacts with a hydrophobic residue of TM3 (Ile$^{3x46}$). In turn, the side chain of the tryptophan "toggle switch" (Trp$^{6x48}$) rotates and TM6 bends at the level of the Phe$^{6x44}$ residue, forming an interaction between the Ile$^{3x40}$ and Phe$^{6x44}$ residues of the P$^{5x50}$I$^{3x40}$F$^{6x44}$ motif. Furthermore, agonists can create a hydrogen bond with Ser$^{5x46}$ resulting in an inward bulge in TM5. [33] Interestingly, this serine is substituted by an alanine residue in both rodent 5-HT$_{2A}$ receptors and human 5-HT$_{2B/C}$ receptors. [34] Therefore, this residue is thought to be involved in differences in

PLOS Computational Biology

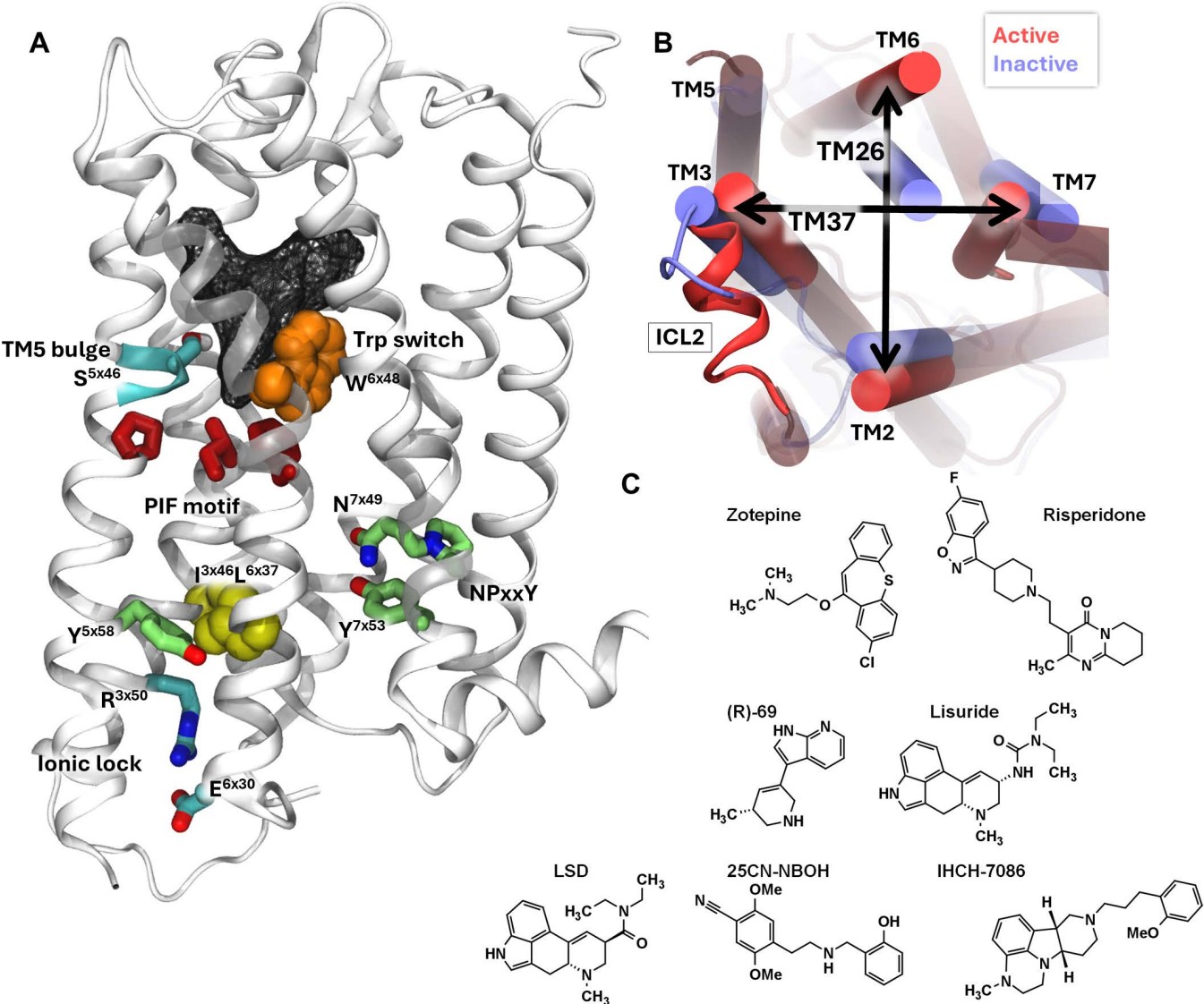

**Fig 1. Important features and ligands in the available experimental structures of the 5-HT$_{2A}$R.** (A) The serotonin-2A receptor with important side chains highlighted. (B) The intracellular side of 5-HT$_{2A}$R. In red the conformation in its active form and in blue its inactive one. The two arrows represent the intracellular helix distances between TM2-6 and TM3-7, respectively. Two important distances to discriminate between states (C) An overview of the ligands used for MD simulations.

ligand affinity through different species, sub-type selectivity binding, binding kinetics [35] and biased signaling. [36] It should be mentioned that these microswitches can be toggled independently, i.e., that ligands can alter the energetics for each switch differently. [37]

Throughout the GPCR family, the highly conserved tyrosines Tyr$^{5x58}$ and Tyr$^{7x53}$ (the latter is part of the NPxxY motif) forms a mediated hydrogen network which is believed to stabilize the active state. [38] More specifically for 5-HT$_{2A/B/C}$ receptors, residue F329$^{6x41}$ stabilizes an alternative rotamer of Y254$^{5x58}$ through π-π stacking whereas this residue in other GPCRs contains a hydrophobic non-aromatic side-chain.[36]

## Modulation of 5-HT$_{2A}$

Previous studies [33,39–42] have shown that multiple combinations of "on" and "off" microswitches are likely relevant in the activation pathway and/or the differential response to different types of GPCR ligands. To survey the resulting landscape of states in the specific case of the serotonin-2AR, we performed MD simulations with 7 different ligands; zotepine, an antipsychotic antagonist, risperidone, an antipsychotic inverse agonist, IHCH-7086, a β-arrestin biased partial agonist, (R)-69, a G-protein biased partial agonist (both with retained hallucinogenic properties), lisuride, a G-protein biased partial agonist, and lastly, 25CN-NBOH and LSD, two β-arrestin biased partial agonist with hallucinogenic properties. These structures are displayed in Fig 1C. As it became clear during the study that an intracellular binding partner is necessary for the receptor to reach its active state, we performed each MD simulation in the absence and presence of an intracellular G protein construct. This allowed us to correlate the various pharmacological profiles of the selected ligands with the dynamics of the 5-HT$_{2A}$R by mapping their different effects on key microswitches as well as on large-scale helix movements during MD simulations.

## 2. Materials and methods

Seven distinct experimental structures of the 5-HT$_{2A}$R, each complexed with ligands exhibiting different pharmacological effects, were used to explore the conformational landscape by means of atomistic molecular dynamics (MD) simulations. Fifteen simulations were performed using these templates. Table 1 provides an overview of these simulations along with the pharmacological properties of the ligands.

### System preparation

The X-ray and cryo-EM structures listed in Table 1 were downloaded from the RCSB PDB databank (http://www.rcsb.org/pdb). [43] and prepared for MD using CHARMM-GUI. [44]

***Preparation X-ray coordinates.*** PDB IDs 6A93 and 6A94 represent a crystallographic dimer, from which only chain A was retained. In PDB IDs 6A93, 6A94, 7WC7, and 7WC9, the highly flexible intracellular loop 3 (ICL3) was replaced with the rigid apocytochrome b562RIL for crystallographic purposes. This fusion protein may affect the receptor's conformational equilibria and was therefore removed. [45] This resulted in a chain break between residues 265 (in TM5, the 5th transmembrane helix) and 313 (TM6). The CHARMM-GUI PDB manipulator [46–48] was used to cap the N- and C-termini of the protein - including the ICL3 chain break - with acetyl and methylamide groups, respectively. During this step, the mutations S372N, S162K, and M164W were reversed. Disulfide bonds were defined between Cys148-Cys227 and Cys349-Cys353. CHARMM-GUI's loop modeling functionality was employed to model the unresolved residues 181–187 (ICL2) in 7WC7 and 7WC9. In addition, the PDB of 7WC9 included 6 monoolein molecules. Accordingly, a topology file for monoolein was constructed manually from CHARMM lipid force field [49] atom types; see "available data".

***Preparation cryo-EM coordinates.*** The unresolved residues 100–104 (ICL1), 143–144 (ECL1), 224 (ECL2), and 394–401 (helix 8) in PDB IDs 6WHA and 9AS4 were modeled using CHARMM-GUI's loop modeling functionality. Conversely, TM1 and extracellular loop 3 (ECL3) were not fully resolved in the cryo-EM structures, but the modeling for these regions resulted in unrealistically strained structures. Instead, for 6WHA, the coordinates of those amino acids (TM1; 71–83 and ECL3; 348–353) were copied from 7WC9 after alignment of the surrounding secondary structure elements. When the structures of 8UWL, 7RAN, and 9AS4 became available, a production run of more than 1,500 ns using 6WHA was already in progress. RMSD time series were calculated for the 6WHA trajectory, using the 8UWL, 7RAN, and 9AS4 cryo-EM structures as references. To improve model accuracy, the frame with the lowest RMSD value for each system was selected as the source for filling in its missing coordinates of TM1 and ECL3 (following structural alignment). Disulfide bonds between Cys148–Cys227 and Cys349–Cys353 were again defined.

**Table 1. Overview of the molecular dynamics (MD) simulations performed and the ligands used.** Simulations are grouped as non-G-protein coupled (i.e., without the G protein construct; simulations 1–8) and G-protein-coupled (with the G protein construct; simulations 9–15). The PDB-template structures from which each simulation was initiated are also indicated and color-coded according to their inactive (blue) or active (red) receptor state.

| Ligand | pharmacological effect | Non-G-protein coupled systems | | | G-protein coupled systems | | |
|---|---|---|---|---|---|---|---|
| | | Sim no | PDB entry template | Time (µs) | Sim no | PDB entry template | Time (µs) |
| Zotepine | Antagonist (antipsychotic) | 1 | 6A94 inactive | 2.8 | 9 | 6WHA active | a) 1.9 b) 1.8 |
| Apo | | 2a | 6A94 inactive | 4.8 | 10 | 6WHA active | a) 2.4 b) 2.1 |
| | | 2b | 6WHA active | 4.1 | | | |
| Risperidone | Inverse agonist (antipsychotic) | 3 | 6A93 inactive | 3.2 | | | |
| Lisuride | G-protein biased partial agonist (non-psychedelic) | 4 | 7WC7 inactive | a) 1.4 b) 1.6 | 11 | 8UWL active | a) 2.1 b) 1.3 |
| (R)-69 | G-protein biased partial agonist (non-psychedelic) | 5 | 7RAN active | a) 3.4 b) 3.6 | 12 | 7RAN Active | a) 1.8 b) 1.2 |
| IHCH-7086 | β-arrestin biased partial agonist (non-psychedelic) | 6 | 7WC9 inactive | 4.2 | 13 | 6WHA active | a) 1.1 b) 1.0 |
| 25CN-NBOH | β-arrestin biased partial agonist (psychedelic) | 7 | 6WHA[b] active | 3.0 | 14 | 6WHA active | a) 1.9 b) 1.8 |
| LSD | β-arrestin biased partial agonist (psychedelic) | 8 | 9AS4 active | a) 1.5 b) 1.3 | 15 | 9AS4 active | a) 2.4 b) 2.1 |

The N- and C-termini, including those at the ICL3 chain break, were capped with acetyl and methylamide groups, respectively. The cryo-EM structures also contained a mini-Gq protein [50] complexed with a single-chain variable fragment (scFv16), both of which were initially excluded during system preparation.

*Simulation box.* The "preprocessed" proteins resulting from the above procedure were solvated in a rectangular periodic membrane system using CHARMM-GUI [49,51–56] The box dimensions and water layer thickness were selected to ensure that the minimum distance between the receptor and its periodic images was at least 24 Å. The receptor's orientation within the membrane was determined based on the Orientations of Proteins in Membranes (OPM) database. [57] Ultimately, the X and Y dimensions of the box were both set to 75 Å and the water thickness to 13.5 Å. Next, a heterogeneous bilipid membrane layer was selected which represents the biological environment of the receptor. Specifically, based on the lipid composition of synaptic vesicles, [58] the membrane was built with 57 POPC (1-palmitoyl-2-oleoyl-sn-glycero-3-phosphocholine), 18 POPS (1-palmitoyl-2-oleoyl-sn-glycero-3-phospho-L-serine), 8 PSM (1-palmitoyl-2-stearoyl-sn-glycero-3-phosphoethanolamine), 12 CER160 (ceramide 160), 52 POPE (1-palmitoyl-2-oleoyl-sn-glycero-3-phosphoethanolamine), 4 POPI (1-palmitoyl-2-oleoyl-sn-glycero-3-phosphoinositol) and 42 cholesterol molecules. The replacement method was used to build the membrane. After performing a ring penetration check, the system was solvated in a 0.15 M KCl solution using the Distance Monte Carlo algorithm implemented in CHARMM-GUI.

*G protein construct.* For simulations 9–15, a small construct representing the portions of the G protein that influence the receptor was assembled by modifying the PDB files of 6WHA and 7RAN. Specifically, residues 25–39, 69–86, and 222–246 of the mini-Gq protein in chain B were retained at their original positions. All N- and C-termini resulting from breaking

the chains were capped with acetylamide and methylamide groups, respectively; note that this does not include the actual C-terminus of the chain as used in the cryo-EM structure (residue 246). The retained residues are shown in S1 Fig. To preserve the geometry of the original mini-Gq protein, distance restraints were applied to a selected set of atoms (listed in S1 Table along with their corresponding force constants). After positioning the resulting construct next to the receptor based on structure alignment, the proteins were solvated in rectangular periodic membrane systems as described above (but resulting in a slightly larger box). Time series of the Z-distance between the helical segment of the G protein that interacts with the receptor and the receptor itself are presented in S2 Fig. The angle between this helix and the xy-plane is also shown. These plots indicate that no decoupling of the G protein construct ever occurred. In addition, ICL2, which adopts a helical conformation in the active state of 5-HT$_{2A}$, remained helical throughout these simulations. By contrast, in simulations where the G protein construct was removed, ICL2 transitioned to a disordered loop and TM6 shifted toward its inactive conformation within the first few hundred nanoseconds after the simulation began. These observations suggest that the G protein construct helps stabilize key structural elements of the receptor's active state.

*Ligands.* The mol2 files of zotepine, risperidone, lisuride, IHCH-7086, (R)-69, 25CN-NBOH, and LSD were generated by building the ligands in Avogadro, followed by protonation at their most basic nitrogen atom using OpenBabel [59]. For all ligands, this resulted in protonation of the basic amine, which is important because experimental structures suggest a polar hydrogen bond with D155$^{3x32}$ of the receptor. Topology and parameter files for each ligand were automatically generated with CGenFF, based on the mol2 files, during the PDB manipulation step [51] Simulations 2b and 10 were initiated from the 6WHA template, with 25CN-NBOH removed. In simulations 9 and 13, 25CN-NBOH was replaced with zotepine and IHCH-7086, respectively.

## Equilibration steps and molecular dynamics

The system was heated and simulated with NAMD 2.14 [60] using the standard protocol and simulation parameters from CHARMM-GUI. [49,61] Briefly, a 10000-step minimization was followed by three 250 ps equilibration steps and three more 500 ps equilibration steps, each at 303.15 K but with gradually weakening restraints. In the subsequent production phase, a time step of 2 fs was used in conjunction with restraints on the hydrogen atoms using the SHAKE algorithm. The pressure of the system was kept at 1 atm using NAMD's Nose-Hoover Langevin-piston barostat with a piston period of 50 fs and a piston decay time of 25 fs. A constant temperature was maintained using Langevin dynamics with a damping coefficient of 1 ps$^{-1}$. Non-bonded interactions were smoothly switched off over 10–12 Å using a switching function for the electrostatics and a force switching function for the van der Waals interactions (as recommended for using the CHARMM force field in NAMD). The long-range electrostatic interactions were calculated using the particle-mesh Ewald method with a mesh size of 1 Å and a 6th order spline interpolation.

Input files and trajectories were deposited on the Zenodo repository (see "Data availability statement".

## Trajectory analysis

Trajectory visualization and alignment were performed using VMD 1.9.3. [62] Measurements of the degrees of freedom (as defined in S2 Table) and RMSD calculations (see timeseries in S3-S4 Figs) were carried out using MDAnalysis 2.4.2. [63] Principal component analysis (PCA) was applied to the time series of the degrees of freedom (DOF) using the Scikit-learn Python package. [64] For each system, 3,000 frames were evenly sampled from the trajectory before performing PCA. To improve visualization, the first part of the simulation—where the system had not yet reached a stable state—was removed based on the RMSD data. S3 Table shows which frames were discarded for each system. Histograms, 2D probability density plots, and contour plots were generated using SciPy's kernel density estimator.[65] Bin widths were set to 0.2 Å and 1º for distances and (dihedral) angles, respectively. The 2D probability density plots from the different MD simulations were Boltzmann inverted but these populations densities should not be interpreted as converged free energy profiles; the logarithm merely makes the wide range of population densities more manageable as well as expressing them

in more intuitive units of energy. For these plots, the initial portion of the trajectory was excluded to ensure that only equilibrated regions of the simulation were visualized; the discarded frames are as well listed in S3 Table.

## 3. Results and discussion

### 3.1 Principal component analysis

Given the wide range of microscopic modulations of the 5-HT$_{2A}$ receptor in response to various ligands, we attempt to systematize the changes in the aforementioned macro- and microswitches. Known relevant DOF in the receptor are listed in S2 Table, along with their definitions and measurement methods. To find patterns in the high-dimensional space of these DOF, it was subjected to PCA, retaining the first two principal components (PCs) for closer interpretation. [64] Most important DOFs are displayed in Fig 1.

When considering the PCA loadings plot (Fig 2A-B), the hydrophobic interaction (I$^{3x46}$ L$^{6x37}$), ionic lock and the attributes related to the movement of TM6 are all situated along the positive axis of in PC1, indicating that they constitute the major source of correlated attributes. In addition, the helix distances related to TM3 and TM7 appear along the negative PC1 axis, indicating an anticorrelation. This is in excellent agreement with the consensus activation mechanism wherein they play important roles, which suggests that PC1 distinguishes between inactive and active states. This is confirmed by the RMSD values (including those of the NPxxY motif) calculated relative to the inactive and active conformations, which correlate and anticorrelate with the aforementioned degrees of freedom, respectively.

The trajectories of the MD simulations listed in Table 1 are represented by the two PCs in Fig 2C, where the circles and triangles represent the receptor's X-ray and CryoEM state, respectively, while the regions of dots and crosses represent frames of different trajectories. It is readily apparent from the figure that the simulations without G protein are clustered on the left while the simulations with G protein are clustered on the right of the plot. This is a first indication that the presence of the G protein may be necessary for the receptor to reach its fully active conformation (as has been shown for β$_2$ adrenoreceptor by Dror and colleagues [40]) More evidence and further discussion regarding this possibility are presented below.

Intriguingly, when examining the G protein bound simulations in Fig 2C, it would appear that positive values of PC2 are associated with the activating/hallucinogenic character of 25CN-NBOH (cyan, simulation 14) and LSD (yellow, simulation 15), whereas the apo structure (green) and the antagonist zotepine (red) have the most negative PC2 values; lisuride, (R)-69, IHCH-7086 are situated in-between (brown, purple and blue, respectively). The DOFs related to the NPxxY motif are a major contributor to the variance along PC2. RMSD time series, calculated relative to their initial geometries for all trajectories, are provided in S3-S4 Figs for both the receptor and the NPxxY motif specifically. In the G protein–bound simulations (S4 Fig), the NPxxY motif maintains a low RMSD during the 25CN-NBOH and LDS simulations. For the non-hallucinogenic ligand IHCH7086, (like the psychedelics, a β-arrestin-biased agonist), an equilibrium appears to exist between low and high-RMSD conformations of the NPxxY motif. This may explain the two population distributions (blue crosses) observed in Fig 2C. During the simulations of the G protein–bound apo form (10) and with the G protein-biased ligands (R)-69 (12), and lisuride (11a), the NPxxY motif gradually transitions to a high-RMSD state within the first 200–300 ns. These observations might suggest that NPxxY is indicative of β-arrestin bias, or alternatively, that both a strong activating ligand and an intracellular binding partner are necessary to maintain the NPxxY motif in an active-like conformation (the distinction between the two possibilities is somewhat ambiguous in the present data set).

### 3.2 Removal of the G protein

Time scales in MD simulations are generally too short to reliably observe the transition of 5-HT$_{2A}$R to an active state. However, studies have shown that an intracellular binding partner is necessary for at least some GPCRs to maintain their active conformation. [38,66] By removing the G protein, we can study the *deactivation* mechanism instead. In this context, Dror and colleagues [40] observed three distinct states during the MD simulation of β$_2$ adrenoreceptor in which an

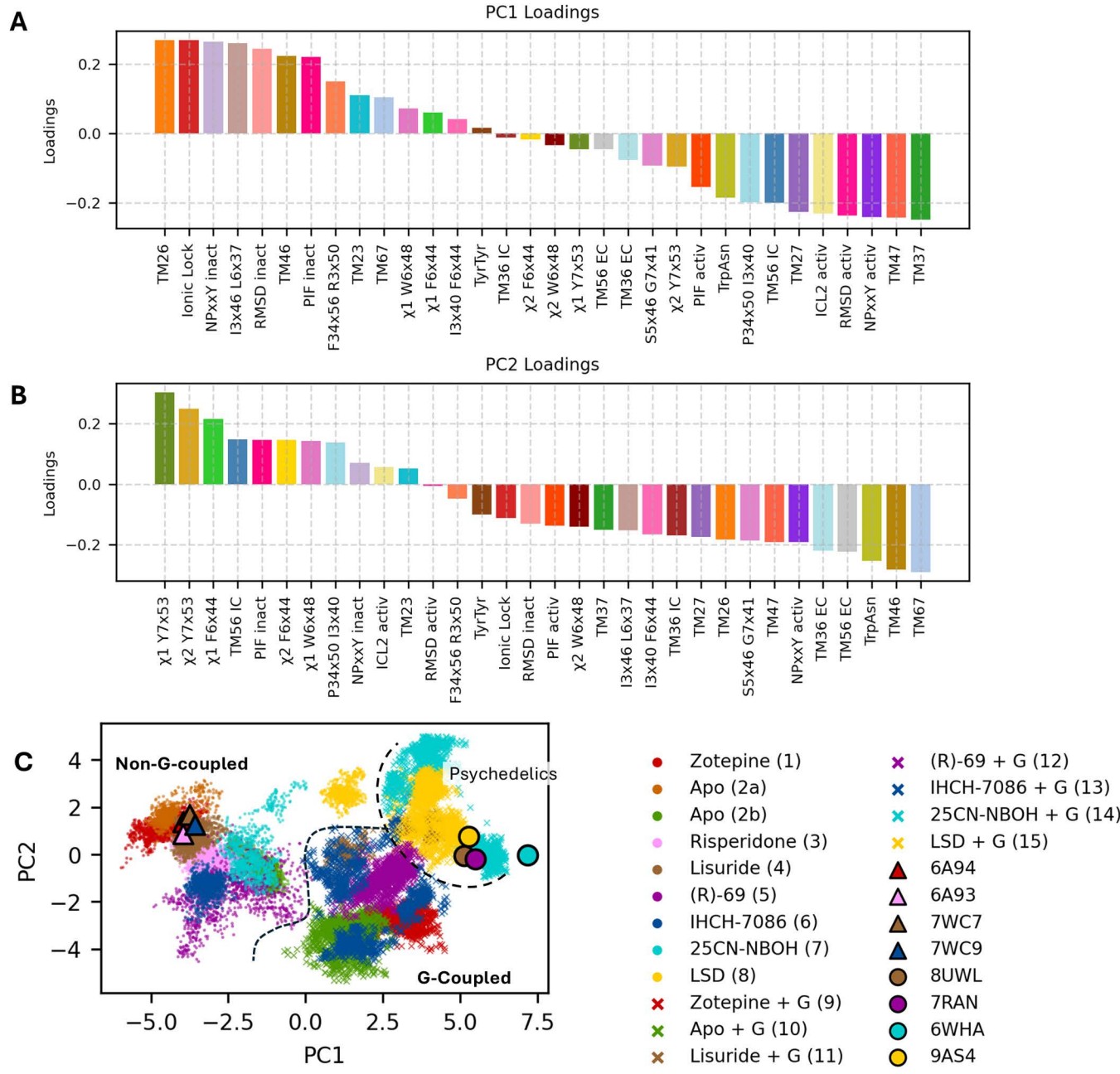

**Fig 2. Results of the PCA analysis.** (A–B) Loadings for each DOF (as defined in **S2 Table**) for the first two principal components. The eigenvalues of the covariance matrix are 10.8 and 3.9, respectively, with explained variance ratios of 34% and 12% for PC1 and PC2. (C) Projection of simulations 1–15 and corresponding X-ray/cryo-EM structures onto the first two principal components.

intracellular nanobody stabilizing the receptor's active state was removed. Specifically, a plot of the RMSD of the NPxxY motif of an inactive conformation versus the TM3-TM6 distance showed three distinct clusters of conformations. The first cluster is positioned around the crystal structure. A second cluster is formed after the rearrangement of the NPxxY motif near the end of TM7. From this state, a last cluster is obtained when the distance between TM3 and 6 decreases. To

replicate the observations leading to this non-canonical activation pathway for the 5-HT$_{2A}$R, similar plots were constructed in S3 Fig for simulations 2b, 5 and 7, in which the mini-G$_q$ protein was removed from their corresponding PDB templates. In S5 Fig, the black star on the 2D plot represents the cryo-EM structure. In the first row are the TM3–6 and the RMSD of the NPxxY values plotted, revealing two clusters: one with a high RMSD value and with low RMSD values. Overall, these plots suggest that the NPxxY motif initially shifts toward a more inactive conformation, followed by a decrease in the TM3–6 distance before it subsequently increases again. An exception occurs in simulations 5a and b with ligand (R)-69 after the removal of the G protein. This simulation was run twice and transitioned to an inactive or partially active conformation along two distinct pathways. This would seem to imply that the rearrangements of the helices do not necessarily have to happen in a specific order.

Fig 3 shows how the TM2–6 distance and the RMSD of the NPxxY motif relative to the inactive state evolve during simulations 2b, 5 and 7 and their corresponding G protein-coupled counterparts (10, 12 and 14) as reference. As foreshadowed by the PCA, it becomes evident that the presence of the G protein construct prevents inward movement of TM6. However, it does not seems to prevent the NPxxY motif to assume an inactive-like conformation for the apo and (R)-69 simulation, indicating the ligand's ability to modulate this structural motif.

The same analysis was also performed for two other degrees of freedom: the intracellular TM2–6 and TM3–7 distances, as defined in S2 Table. This pair of distances showed a large anticorrelation in the PCA (Fig 2) and was chosen as representative for the shape of the intracellular binding surface. Indeed, as shown in Fig 1B, helices 2, 3, 6 and 7 form a square delineating this binding surface. Combining the information from the different experimental structures, a global anticorrelated motion must occur along the perpendicular TM6–2 and TM3–7 axes during the transition from an inactive apo state to a fully active state containing the mini-Gq protein and an activating ligand. However, our MD simulations

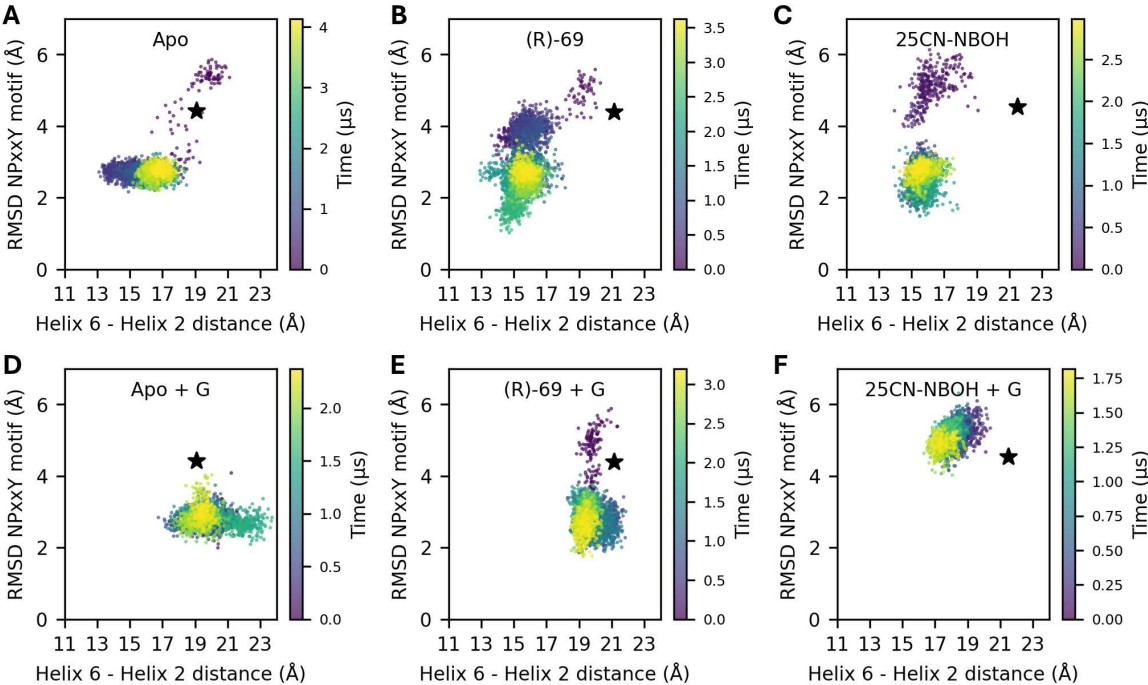

**Fig 3. Time series showing the TM2–6 distance and the RMSD of the NPxxY motif relative to the inactive state.** Panels (A–C) present trajectories from simulations 2b, 5, and 7. Panels (D–F) show the corresponding trajectories for G protein–coupled systems from simulations 10, 12, and 14.

indicate that two motions tend to happen sequentially rather than in a concerted fashion, as can be seen in S5H-J Fig. These motions are explained in more detail in § 3.4.

### 3.3. Microswitches, structural motifs and side chain conformations

The outward movement of TM6 is often associated with a bend in the helix near residues W336$^{6x48}$ and F332$^{6x44}$. As suggested by Wallach and colleagues [30] and Liao and colleagues [41], different rotamers of W336$^{6x48}$ may play a role in biased signal transduction. Their studies indicate that when the switch undergoes a further downward movement in the binding pocket, corresponding to smaller $\chi_2$ values, it promotes β-arrestin transduction. More recently, this was experimentally confirmed by Gumpper and colleagues [67] with the β-arrestin biased ligand RS130–180. Fig 4 displays the $\chi_1$ and $\chi_2$-angles of these residues. Consistent with their studies, the 25CN-NBOH simulations (7 and 14; cyan) display a transition in the $\chi_2$ angle of the tryptophan toggle switch between ~120° ("off" state) and ~50° ("on" state) (Fig 4A). Additionally, the presence of the G protein promotes the "on" state in both the apo simulation (8) and the 25CN-NBOH–bound simulations (11).

However, Fig 4A reveals an unusual transition of W336$^{6x48}$ in the (R)-69 bound simulation (5b). Instead of moving further down the binding pocket, the indole moiety of W336$^{6x48}$ rotates away so that $\chi_1$ assumes values between 30–60º (plotted in Fig 4A, and side chains illustrated in Fig 4C). This specific indole conformation is only observed in simulations with the G protein biased ligand (R)-69 and in the receptor-G protein complex in combination with a potent agonist like 25CN-NBOH. Therefore, this specific state depicted in Fig 4C could be an interesting target for future docking studies aimed at identifying additional G protein-biased ligands with low efficacies.

Additionally, Fig 4B displays $\chi_1$-$\chi_2$-plots for F332$^{6x44}$, which is part of the highly conserved PIF motif. The $\chi_1$-value indicate that the benzyl moiety can adopt a gauche-like and antiperiplanar-like conformations (Fig 4D). Notably, the antiperiplanar conformation is observed only in the presence of strong activating ligands (i.e., simulations 7, 8, 14 and 15) or intracellular binding of mini-Gq (as in simulation 10). While in this anti conformation, the PIF-motif assumes an "active"-like conformation, this effect might initiate a cascade of conformational changes throughout the receptor's intracellular side, biasing it toward binding an intracellular partner. The agonist (R)-69-bound system (simulation 5; purple) jumps on a very few occasions also to this anti state while its Trp-switch assumes this "alternative state. Consequently, the $\chi_2$ angle distribution of F332$^{6x44}$ increases, as observed in simulations 10, 14 and 11, suggesting that the phenyl-moiety of F332$^{6x44}$ obtains a higher mobility. In turn, this might as well lead to intracellular signaling. Conversely, the antagonist zotepine penetrates much deeper into the binding pocket, blocking the tryptophan toggle switch and preventing interactions between the side chains in the connector region (PIF motif) (Fig 4E).

These observations might help explain the pharmacological differences between the (R)-69 and 25CN-NBOH agonists. Although they occupy roughly the same binding pocket, 25CN-NBOH possesses an additional aromatic ring which seems to be incompatible with the inactive state of the tryptophan switch, thus forcing it into its "on" state, whereas (R)-69 biases the tryptophan residue to an alternative "on" position from where its inactive state remains accessible (Fig 5A-5B).

For the N-benzylated phenethylamine, 25CN-NBOH, it was hypothesized that pushing the W336$^{6x48}$ toggle switch was essential for its potency and efficacy. [30,36,67] Indeed, Fig 5B shows that as the indole ring is pushed downward, a hydrogen bond forms between W336$^{6x46}$ and N376$^{7x49}$ of the NPxxY motif. This interaction stabilizes the active, "kinked" helix of TM7, which may explain the consistently active NPxxY conformation observed during simulations 14. This hydrogen bond is not observed in the other simulations as can be seen from the distance distribution between W336$^{6x46}$ and N376$^{7x49}$ in Fig 5C. However, while this H-bond might further stabilize the active NPxxY state, we could observe for the RMSD timeseries of IHCH-7086 and LSD (S4 Fig) that this feature is not strictly necessary to achieve an active-like NPxxY state.

When TM7 adopts its kinked helical structure, the tyrosine residue of the NPxxY motif (Y380$^{7x53}$) shifts inward, corresponding to an active conformation (Fig 5B). As also indicated by the PCA, a shorter distance between Y380$^{7x53}$ and

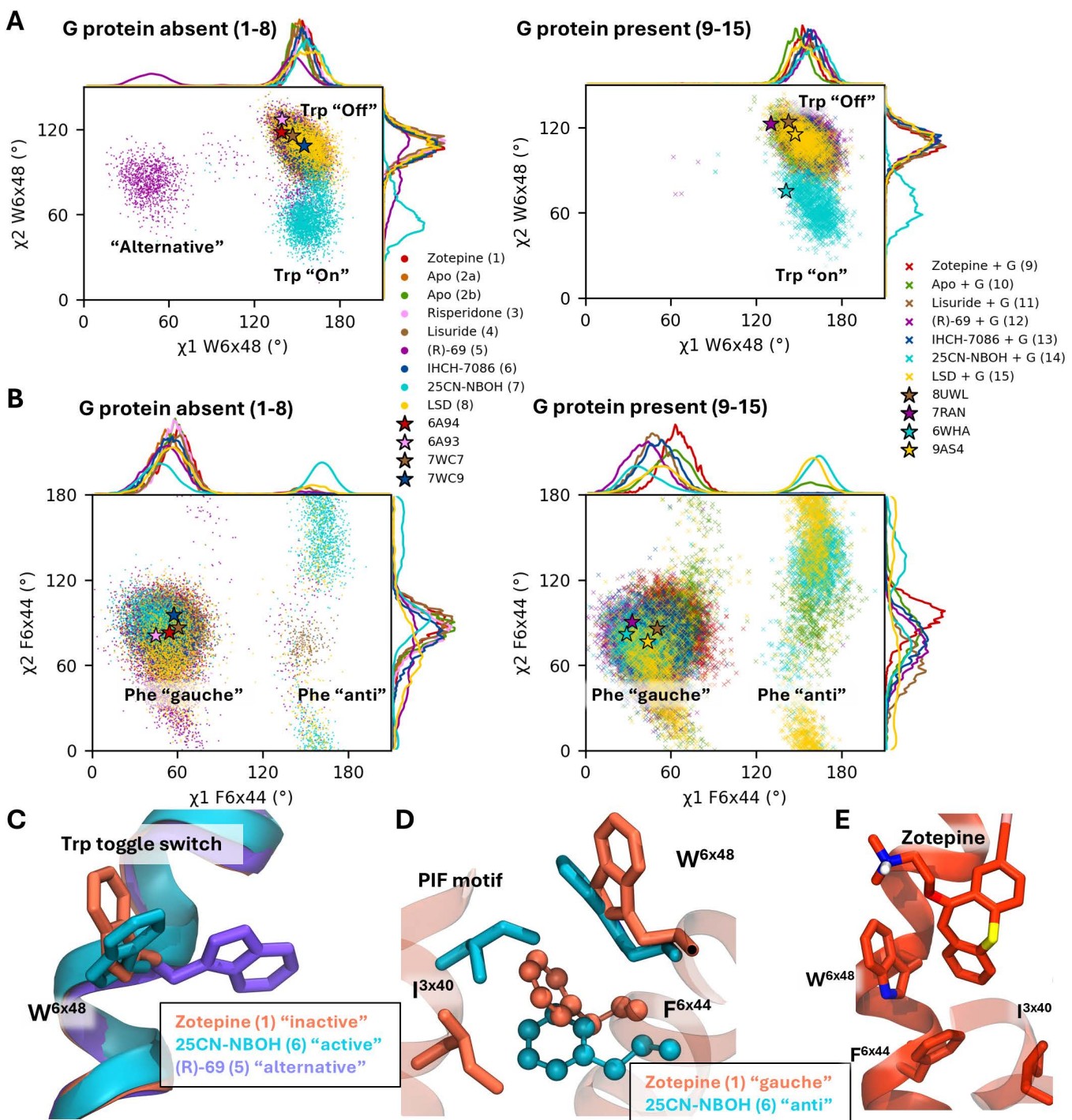

**Fig 4. Analysis of side chain conformations of microswitches.** $\chi_1$ and $\chi_2$ angles of (A) the W336[6x48] switch and (B) the F332[6x44]. (C) Overlap between an "on" tryptophan switch (cyan) with the hallucinogen, an "off" one (orange) with antagonist zotepine and an "alternative" rotamer (purple) with (R)-69. (D) An active and inactive state of the PIF motif with F332[6x44] a gauche and anti-like aconformation. (E) Blocked "Trp-switch" with antagonist zotepine.

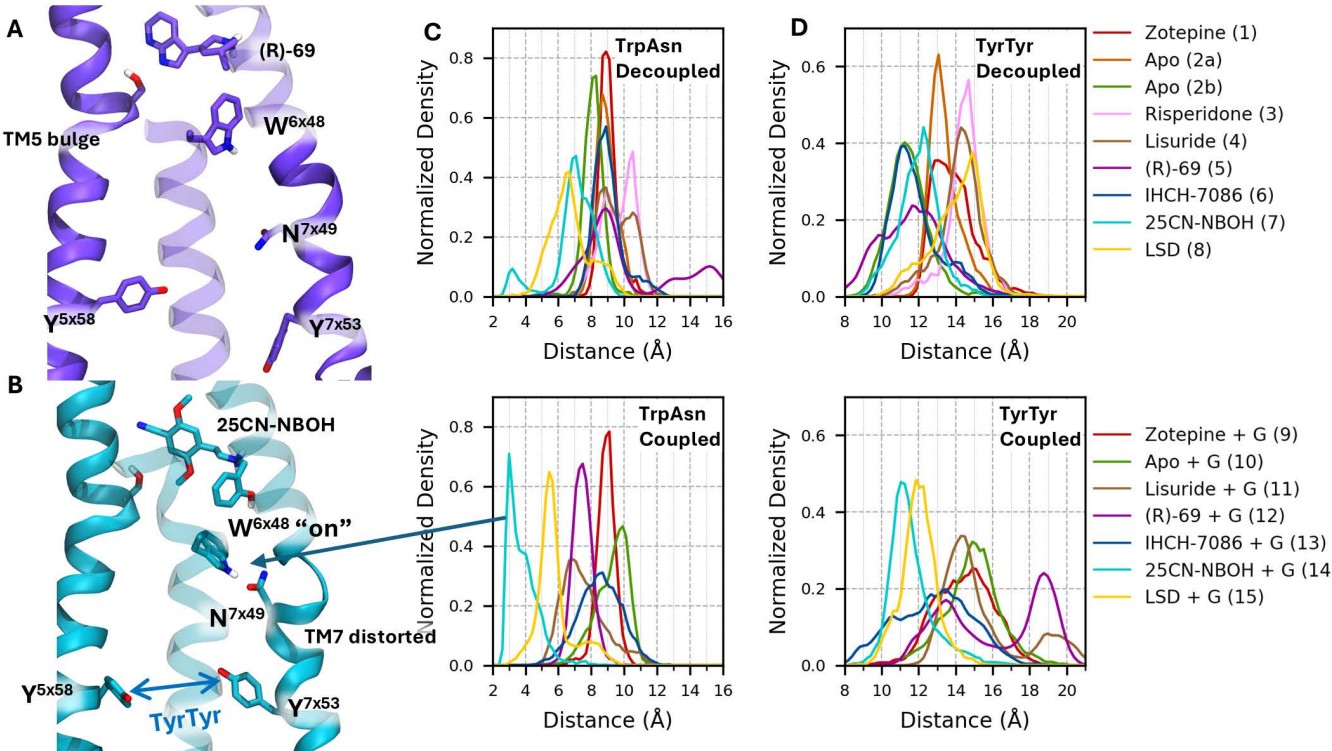

**Fig 5. Analysis of important inter-side chain distances as identified by the PCA.** (A-B) 5-HT$_{2A}$ with (R)-69 and 25CN-NBOH, resp. Normalized histograms are shown for simulations 1-8 without the G protein (top row) and simulations 9-15 with the G protein construct (bottom row). (C) Distance between indole-nitrogen of W336$^{6x48}$ and amide-oxygen of N376$^{7x49}$. (D) Distance between hydroxyl-oxygens of Y254$^{5x58}$ and Y380$^{7x53}$.

Y254$^{5x58}$ (TyrTyr) is associated with a more active conformation. For the $\beta_2$ adrenergic receptor it was proposed that both tyrosines face inward in the active state, stabilizing it by forming a water mediated H-bond network. [33,38,40] However, in the active states of 5-HT$_2$ subfamily, the phenolic group of Y254$^{5x58}$ is displaced outward from TM6 and it interacts with F329$^{6x41}$ by means of π-stacking. This has been suggested to contribute to the preference of the receptor to couple with proteins of the Gq family. [36] During the (R)-69 simulation, an inward shift of Y254$^{5x58}$ was observed while Y380$^{7x53}$ was directed outward (Fig 5A).

To further systematize ligand–receptor interactions, a second PCA analysis was performed, excluding the Apo trajectories from the data set and extending the list of variables with five DOFs (S6 Fig) associated with the ligands. Since the latter do not belong to a congeneric series, the additional DOFs were simply calculated as the shortest distance between a polar non-hydrogen atom of the residue (shown in Fig 6A and 6C) and the nearest polar non-hydrogen atom of the ligand. In this new PCA, the first PC essentially expressed the same degree of freedom as before, i.e., the large-scale displacements associated with the activation mechanism continue to dominate the PCA. In sharp contrast, PC2 is now entirely different, containing important contributions from the TM5 bulge (S242$^{5x46}$), the ligand, and TyrTyr. Notably, (R)-69, lisuride and LSD, the agonists with a pronounced ability for G protein signaling, are the only ligands that form a H-bond with S242$^{5x46}$ during our simulations (see S7 Fig). This interaction leads to a greater distortion of TM5 (the TM5 bulge; see Fig 5A), which explains the shorter S242$^{5x46}$ – G369$^{7x41}$ distances compared to the other ligands shown in Fig 6B. The presence of the intracellular binding partner further reduces these distance distributions, indicating an even greater distortion of TM5 near residue S242$^{5x46}$.

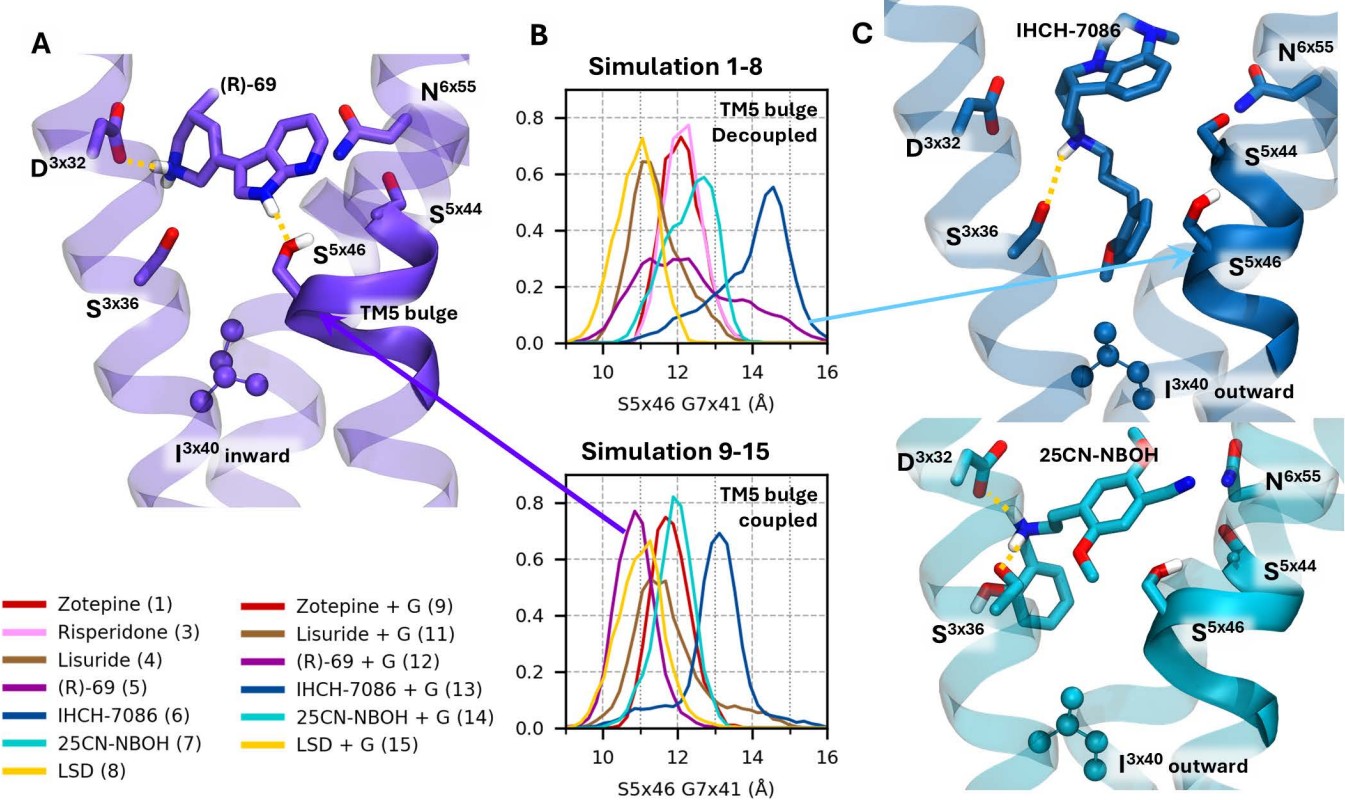

**Fig 6. Analysis of important ligand-side chain interactions.** (A) Structure of 5-HT$_{2A}$ bound to (R)-69, highlighting the five additional DOFs included in the second PCA, along with the TM5 bulge and residue I163$^{3x40}$. (B) TM5 bulge, quantified as the distance between residues S242$^{5x46}$ and G369$^{7x41}$. (C) Structures of 5-HT$_{2A}$ bound to the β-arrestin–biased ligands IHCH-7086 (top) and 25CN-NBOH (bottom).

Like in the agonist-bound β$_2$-ARs, the TM5 "bulge" induces a conformational change in the PIF motif. [33] When S242$^{5x46}$ shifts inward, it moves P246$^{5x50}$ toward I163$^{3x40}$. As a result, the isoleucine residue adopts a different inward rotamer, which subtly influences the position of F332$^{6x44}$. This alteration in the PIF region is thought to modify the energetic landscape of the receptor, facilitating the transition of TMs 6 and 7 from the inactive to the active state.

In contrast to the above, the β-arrestin–biased ligands IHCH-7086 and 25CN-NBOH do not form a H-bond with S242$^{5x46}$ (Fig 6C). As a result, I163$^{3x40}$ adopts a different rotamer. As hypothesized by Cao and colleagues, this difference may underlie their distinct agonist behavior; indeed, following this hypothesis, IHCH-7086 was intentionally designed to form no H-bond with S242$^{5x46}$. The TM5 bulge in simulations 6 and 13 (IHCH-7086) is less pronounced compared to the other simulations, which might explain β-arrestin–mediated signaling without detectable G$_q$ activity. More generally, GPCR signaling bias appears to involve shared allosteric mechanisms, as proposed by Weis and Kobilka. [33] We speculate that the TM5 bulge may influence the PIF motif, thereby communicating with the intracellular side of the receptor to stabilize the active-like NPxxY and Tyr–Tyr motifs. Interestingly, 25CN-NBOH seems to bypass this mechanism by directly pushing the toggle switch W336$^{6x48}$, enabling direct interaction with the NPxxY motif.

### 3.4 Large scale movements

Since the outward motion of TM6 does not happen simultaneously with the inward motions of TM3 and TM7 during activation, different partially-active conformations can be proposed. To further investigate the populations of these different states, 2D probability density plots of TM6–2 versus TM3–7 are displayed in Fig 7.

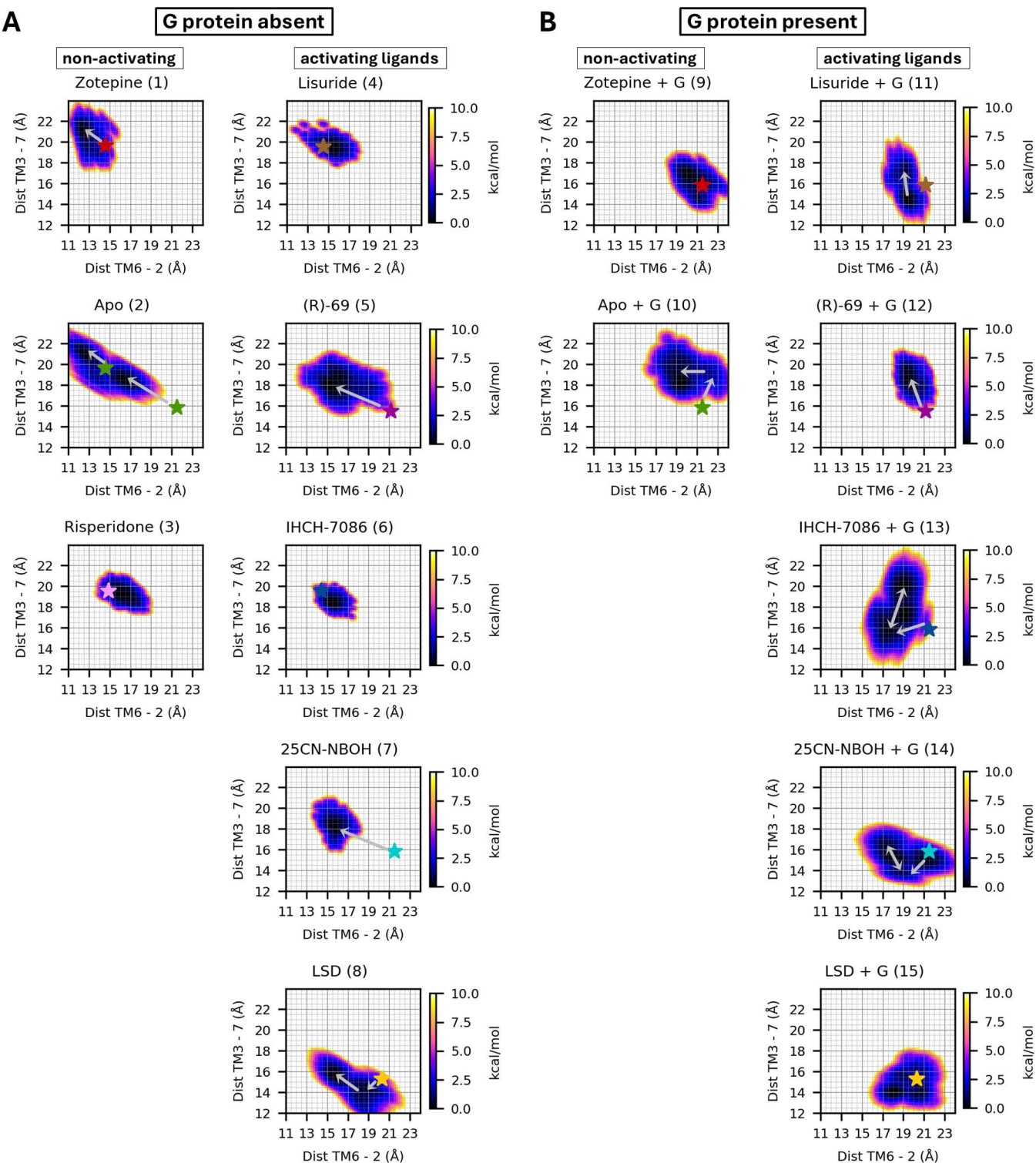

**Fig 7. Boltzmann-inverted density plots for TM6–2 and TM3–7 axes, presented in free-energy units (kcal/mol).** Panel A shows simulations conducted without the G protein construct, while panel B displays simulations with the G protein construct present. Arrows indicate the direction of conformational transitions starting from the experimentally determined structures (denoted by stars).

When comparing the non- and G-bound systems, a notable shift in the density to higher TM26 distances is observed. This shift indicates that the G protein is essential for TM6 to achieve its full outward position, consistent with the PCA results and the discussion in §3.2. In addition, when looking at the plots for the hallucinogen bound receptors (25CN-NBOH; 7 and 14 and LSD; 8 and 14), it can be observed that the minimum is shifted slightly downward, meaning that TM3 and TM7 are shifted inward in response to the presence of the G protein. Combing these observations, it would indeed appear that the hallucinogen-G protein-bound complexes are located the farthest along an activation pathway consisting of an increase in TM2–6 and decrease in TM3–7 distance. Conversely, if we compare the location of this minimum with the other simulations, we can carefully assume that the other G protein-bound complexes are in a partially activated state, since either TM6 is shifted more inward, TM3 and TM7 shifted outward or both. The only exception to this trend is simulation 7, which includes both the antagonist zotepine and the G protein. When comparing the minimum of simulation 7 with those of the non-psychedelic compounds (simulations 10–13), we observe that upon docking zotepine into the G-protein-bound active conformation, the receptor fails to transition to a less-active state over the course of the simulation. This contrasts with other G-protein-bound simulations containing a partial agonist or no small-molecule ligand, where such transitions do occur (indicated by the grey arrows). While the absence of a transition with zotepine is most likely a shortcoming of the computational time constraints, an alternative explanation would be that zotepine hinders the conformational freedom of the receptor. More specifically, it seems possible that zotepine would increase the free energy of one or more intermediary conformations in the activation mechanism, thus inhibiting both deactivation and activation. Indeed, in the absence of the G protein there is a trend away from the upper-left corner of the graph with increasingly activated ligands, while the simulation with zotepine displays a compact distribution corresponding to a fully deactivated state. We further speculate that zotepine could accomplish this by sterically hindering the unraveling of a network of interactions that holds TM6 in place in both the activated and the deactivated conformation. This is supported by Fig 4E, where zotepine hinders the tryptophan switch and the interaction between $F^{6x44}$ and $I^{3x40}$ of the PIF motif.

In apo simulation (10), upon removal of the hallucinogen 25CN-NBOH, the population density shifts toward higher intracellular TM3–7 distances, again indicating that both the G protein and the hallucinogen are necessary for the receptor to remain in the conformational basin corresponding to the active state. In addition, we found that upon G protein activation, the psychedelics 25CN-NBOH and LSD populates a structural different state than the other agonists (Fig 7B). To facilitate comparison across simulations, Fig 8 presents a contour plot of the 50% most frequently sampled TM26 and TM37 distances.

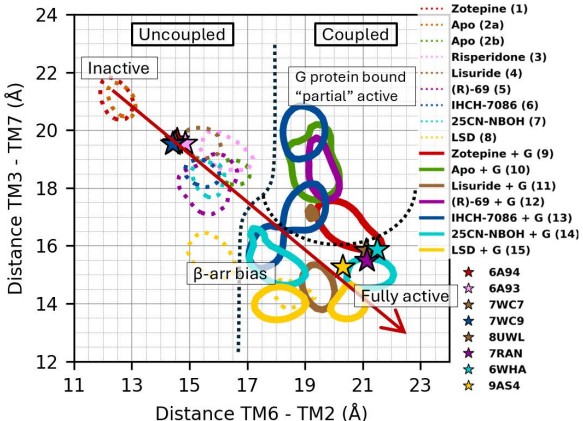

**Fig 8. Contour plot depicting 50% most sampled TM6-2 and TM3-7 distances across simulations 1-15.** Colored stars represents the state their corresponding X-ray or Cryo-EM structures. It shows the distinct conformational regions (labeled by the 5-HT$_{2A}$'s state) modulated by different ligands. A putative "axis of activation" is drawn in red along the TM6-2 and TM3-7 axes.

Comparing the conformational landscapes of the psychedelics (simulation 14 and 15) with the other agonists (11–13) leads us to speculate that the ideal β-arrestin biased receptor conformation would feature a TM37 distance close to 15 Å and a TM26 distance around 18 Å (see Fig 8; blue, cyan, yellow). In this hypothesis, TM26 should be somewhat higher (around 21 Å) for G protein signaling, which would be potentiated by a further decrease in TM37. Combined with 25CN-NBOH's and LSD's high absolute binding affinity for the receptor, this would make it plausible that it causes hallucinations by inducing excessive populations of activated 5-HT$_{2A}$ while at the same time being β-arrestin biased. It would also explain why both IHCH-7086 and (R)-69 are weak agonists while being β-arrestin and G protein biased, respectively.

In summary, after some equilibration, the simulations without G protein largely assumed an inactive state, especially along the TM6−2 axis – however, they showed a clear tendency *in the direction of* the active state for increasingly agonistic ligands, which is most pronounced along the TM3−7 axis as can be observed from the distinguishable regions in Fig 8. Similarly, the simulations with G protein largely remained in an active state (again, especially along the TM6−2 axis) but showed a clear tendency *in the direction of* the inactive state for increasingly inactive or antagonistic ligands, at least along the TM7−3 axis. The only exception to the latter observation is the simulation with zotepine, which biased the conformational ensemble toward an inactive state without G protein, but toward an active conformational ensemble with G protein, despite it being known as an antagonist. This is most likely a consequence of insufficient sampling, as discussed above. Apart from this exception, the global trend is consistent with the hypothesis that an activating ligand by itself does not make 5-HT$_{2A}$ transition to its fully activated conformation, but rather adds a bias to the conformational energetics such that the binding of the G protein *concurrent* with the transition to an active state (presumably near the lower right of Fig 8) becomes energetically more favorable. Indeed, this mode of action has been proposed on the basis of computational and experimental data for the β$_2$-adrenergic receptor [33,40] and might be more widespread in the GPCR family. Going one step further, our results regarding the G protein versus β-arrestin-biased ligands are in excellent agreement with the observation by Weis and Kobilka that despite their differential pharmacology "these [different] agonists likely share allosteric transmission mechanisms". [33] Taken together, this leads us to hypothesize that hallucinations result from excessive populations of activated 5-HT$_{2A}$ and that a non-hallucinogenic antidepressant effect could be achieved by *a sufficiently mild activating ligand*. This may imply an energetic effect that *weakly favors* the "fully activated" conformation (i.e., a *quantitative* effect), either by binding relatively weakly to the receptor (like the endogenous ligand 5-HT, when compared to hallucinogens [68,69]) or by binding strongly but having a modest activating influence of the conformational energetics. Alternatively, a mild activating ligand could induce a strong bias in the receptor's conformational energetics, but toward a "non-ideal" conformation that only weakly enhances the intracellular binding of the G protein- (which we would call a *qualitative* effect). Such a conformation might lie between the inactive and fully active conformation or correspond to the β-arrestin biased conformation (which traditionally would lead to desensitization). Either way, the above distinction between a "qualitative" and "quantitative" effect is difficult to make when studying 5-HT$_{2A}$ in isolation because the conformational difference between the tentative G protein biased and β-arrestin biased conformations is modest, so that Hammond's postulate predicts that both effects will often occur together. This equally implies that this distinction may be of lesser concern in practical efforts to design non-hallucinogenic 5-HT$_{2A}$ antidepressants.

## Conclusions

The results of this study suggest that intracellular binding of the G protein is necessary for the 5-HT$_{2A}$ receptor to assume a fully activated conformation exhibiting the common outward movement of TM6 in GPCRs. Indeed, we propose that the role of activating ligands is not to *directly* induce this outward movement, but to alter the conformational energetics of the receptor such that the (possibly but not necessarily simultaneous) adoption of an activated conformation and intracellular binding of the G protein becomes more favorable, in line with the β$_2$-adrenergic receptor. This variant of the GPCR activation mechanism would naturally give rise to a range of partial agonists that differentially modulate the equilibrium between 5-HT$_{2A}$R's resting and G protein-bound active conformation. We further

hypothesize that, *while strong activators are known to induce hallucinations* [30]*, a sufficiently weak (partial) agonist – either by virtue of weak binding or weak activation upon binding - might be able to avoid this undesirable side effect while still functioning as an effective antidepressant*. It is worth noting that this hypothesis leaves the question open whether such a weak or partial agonist should be G-protein or β-arrestin biased. Indeed, both pathways likely have a shared transmission mechanism at the level of the receptor, as proposed by Weis and Kobilka [33] and supported by the present results. We argue in §3.4 that both intracellular effects will typically occur together because of Hammond's postulate. Therefore, the question becomes purely academic, and is tied to ongoing debates regarding the intracellular signals that give rise to antidepressant effects and which types of 5-HT$_{2A}$ targeted therapies would be effective. It should be noted in the latter context that, if a small population of artificially activated receptors indeed yields an antidepressant effect, then precisely tailored small doses of a full agonist could theoretically be a viable alternative to a suitable partial agonist. While this "microdosing" route has been the subject of considerable interest and debate, [70–72] even if effective, it raises serious pharmacological and regulatory concerns. Overall, our results support the general idea of directing investigative efforts toward (weak) partial agonists, such as (*R*)-69 and IHCH-7086.

Potential future lines of research include developing a similar model for intracellular interactions with β-arrestin and/or attempting to overcome the activation barrier and estimate free energies using enhanced sampling methods and/or studying other receptors such as 5-HT$_{2B/C}$, 5HT$_{1A}$ and TrkB, which may play a role in the specific effects exhibited by such ligands.

## Supporting information

**S1 Table.** Definition of the collective variables (CVs) in order to preserve the geometry of the G protein construct (with atom colors corresponding those in S1 Fig). Each CV is restrained using a flat-bottom harmonic potential, where the third column specifies the range within which no force is applied, and the force constant applied outside this range is provided in the last column.
(DOCX)

**S2 Table.** Attributes used for the PCA. When multiple atoms or a range of residues are selected, the coordinates of the geometric mean of the non-H-atom is used for that DOF. [c] A representative frame of an inactive conformation with zotepine (simulation 1) is used as reference. [d] The cryoEM with pdb entry 6WHA was used as reference.
(DOCX)

**S3 Table. Overview of the initial portions of each simulation (in nanoseconds) that were discarded.** These frames were excluded based on RMSD time series (S3-S4 Figs) to ensure that only equilibrated segments of the trajectories were used for visualization (e.g., PCA and density plots).
(DOCX)

**S1 Fig.** Intracellular side of 5-HT$_{2A}$ (blue and red) with the G protein construct (yellow, silver and orange). For each atom pair to restrain a color was assigned for easy referencing in S1 Table.
(TIF)

**S2 Fig. Time series of the Z-distance between the helical segment of the G protein that interacts with the receptor and the receptor itself, shown for simulations 9–15.** The Z-distance is defined as the difference in the z-coordinate between the centroid of TM2, TM3, TM6, and TM7 (as defined in S2 Table) and the centroid of residues 238–244 of the G protein construct. The Z-angle is the out-of-plane angle between the xy-plane and the vector defined by the centroid of residues 238–244 and the centroid of residues 232–238.
(TIF)

**S3 Fig.  RMSD timeseries of the Cαs of the receptor (blue), the receptor's NPxxY motif (green) and the non-H atoms of the ligand (orange) for the uncoupled simulations (1–8).**
(TIF)

**S4 Fig.  RMSD timeseries of the Cαs of the receptor (blue), the receptor's NPxxY motif (green) and the non-H atoms of the ligand (orange) for the coupled simulations (9–15).**
(TIF)

**S5 Fig.**   Time series for simulations 2b, 5 and 6 represented in (A-C) by the distance TM3–6 and the RMSD of the NPxxY motif *relative to the inactive state*; in (D-F) by the distance TM2–6 and the RMSD of the NPxxY motif and in (H-I) by the distances TM2–6 and TM3–7.
(TIF)

**S6 Fig.  (A–B) Loadings for each DOF (as defined in S2 Table) for the first two principal components including ligand-residue distances.** The eigenvalues of the covariance matrix are 11.9 and 4.3, respectively, with explained variance ratios of 32% and 12% for PC1 and PC2. (C) Projection of simulations 1–15 and corresponding X-ray/cryo-EM structures onto the first two principal components.
(TIF)

**S7 Fig.**   Distance distributions between the closest polar non-hydrogen atom of the ligand and residues S5x46, S5x44, D3x32, S3x36, N6x55.
(TIF)

## Acknowledgments

We thank Master's students Yassine Remissa and Sandra Emmerechts for their contributions to the simulation work.

## Author contributions

**Conceptualization:** Dimitri De Bundel, Kenno Vanommeslaeghe.

**Data curation:** Jordy Peeters.

**Formal analysis:** Jordy Peeters.

**Funding acquisition:** Kenno Vanommeslaeghe.

**Investigation:** Jordy Peeters.

**Methodology:** Jordy Peeters, Kenno Vanommeslaeghe.

**Project administration:** Kenno Vanommeslaeghe.

**Resources:** Kenno Vanommeslaeghe.

**Software:** Jordy Peeters.

**Supervision:** Kenno Vanommeslaeghe.

**Validation:** Jordy Peeters.

**Visualization:** Jordy Peeters.

**Writing – original draft:** Jordy Peeters.

**Writing – review & editing:** Dimitri De Bundel, Kenno Vanommeslaeghe.

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
