## [Decision Letter · Decision Letter 0]

21 Apr 2025

Molecular dynamics study of differential effects of serotonin-2A-receptor (5-HT2AR) modulators

PLOS Computational Biology

Dear Dr. Vanommeslaeghe,

Thank you for submitting your manuscript to PLOS Computational Biology. After careful consideration, we feel that it has merit but does not fully meet PLOS Computational Biology's publication criteria as it currently stands. Therefore, we invite you to submit a revised version of the manuscript that addresses the points raised during the review process.

Please submit your revised manuscript within 60 days Jun 21 2025 11:59PM. If you will need more time than this to complete your revisions, please reply to this message or contact the journal office at ploscompbiol@plos.org. Please include the following items when submitting your revised manuscript:

We look forward to receiving your revised manuscript.

Kind regards,

David van der Spoel

Academic Editor

PLOS Computational Biology

Nir Ben-Tal

Section Editor

PLOS Computational Biology

**Journal Requirements:**

At this stage, the following Authors/Authors require contributions: JORDY PEETERS. Please ensure that the full contributions of each author are acknowledged in the "Add/Edit/Remove Authors" section of our submission form.

2) Please note that the Author Summary should appear in your manuscript between the Abstract (if applicable) and the Introduction, and should be 150-200 words long. The aim should be to make your findings accessible to a wide audience that includes both scientists and non-scientists. Sample summaries can be found on our website under Submission Guidelines:

**Reviewers' comments:**

Reviewer's Responses to Questions

Reviewer #1: The present article by Peeters et al questions the role of hallucinations in the antidepressant-like effect of 5-HT2A receptor agonists and the respective contribution of separate signals associated with G protein Gq and β-arrestin binding. This work is part of a concept suggesting that partial agonists with low efficacies can give rise to antidepressant effects without psychedelic properties regardless of their activation pathway. However, this needs to be confirmed by new studies, on different scales, and it is in this context that this work takes place. One of the main messages delivered herein is that “hallucinations result from excessive populations of activated 5-HT2A and that a non-hallucinogenic antidepressant effect could be achieved by a sufficiently mild activating ligand. This may imply an energetic effect that weakly favors the “fully activated”.

The work and hypotheses raised are interesting and paves the way for a fascinating debate. My (non-specialist) questions concern more integrated concepts, considering the fact that these 5-HT2A receptor ligands are injected in living individuals. I encourage authors to consider them to enrich their discussion.

1) This study focuses on the 5-HT2A receptor but the pharmacological ligand currently available and intended for in vivo administration (either in humans or in animals) also bind and activate other serotonergic (e.g. 5-HT2B/2C – 5-HT1A receptor type) and non-serotonergic receptors (e.g. TrkB). To what extent these elements participate in reinforcing, or on the contrary, attenuating the 5-HT2A receptor signalling pathways should be taken into account and discussed here.

2) Can we imagine genetic mutation in specific domain of the 5-HT2A receptor in human that could contribute to modify the pscychopharmacological activity of 5-HT2A receptors in terms of hallucinogenic and/or antidepressant effects? Is there any evidence to support this view in humans?

3) Minor points

Line49 : it would be interesting to provide here the affinity of these compounds towards the 5-HT2AR, both in human and rodent.

Line 51: replace the term “pharmaceutical activation” by “pharmacological activation”

Reviewer #2: Summary

In this work, Peeters et al. describe a molecular dynamics study of the 5-HT2A receptor in complexes with ligands with diverse pharmacological profiles, ranging from the inverse agonist ritanserin, neutral antagonist zotepine to partial and full agonists 25CN-NBOH, IHCH-7086, and (R)-69, aiming to find structural features that could be linked to distinct pharmacological profiles. Using PCA analysis on a series of receptor descriptors, the authors highlight important features and descriptors potentially associated with the distinct ligand modalities. The authors also identify a pair of intracellular distances as key descriptors for the modality and pharmacological profiles of compounds.

Despite the relevance of such a study in the context of the renaissance of psychedelics as potential pharmacotherapies, the poor preparation of model systems, lack of simulation replicas, lack of statistical analysis, and absence of control/validation modeling experiments jeopardize the manuscript’s results and conclusions. These methodological concerns need to be addressed by the authors before this manuscript can be considered for publication in PLOS Computational Biology.

Major concerns

- The computational experiments are poorly designed. Little attention was given to the preparation of proteins. The experimental 5-HT2A structures have different sequence lengths and different missing domains (including loops and portions of TMs, like the missing TM1 in PDBs 6WHA and 7RAN). It is not clear from the manuscript how these differences were addressed to make the results comparable across simulations and ligand-receptor complexes. Have optimized models been built, ensuring that the receptor have the same sequence length for all systems? Else, how can the results be comparable?

- Where were the long N- and C-termini truncated? More importantly, what treatment was given to the missing ECL3? Has it been left truncated/open, or has it been closed via some sort of pseudo-loop? These different treatments for the ECL3 can strongly affect the intracellular distances between TMs, from which most of the conclusions are derived, and yet, this is not properly described and apparently this treatment if left to an automated tool to address (Charmm GUI’s PDB manipulator) with little control over the output and admittedly a “rudimentary” tool (sic. L138).

- It is also not correct to “copy” the coordinates from TM1 from an inactive state structure (PDB 7WC9) into the active state receptor. A proper homology modeling tool (e.g., Modeller, Swiss-Model, etc.) should have been used for template-based modeling of the missing TM, which would take into account the neighboring atoms when building the side chains, with interactions and repulsion properly addressed via molecular mechanism and a force field.

- What treatment was given to the sodium binding site in the vicinity of D1202.50? Has this residue’s protonation state been properly adjusted to reflect the receptor state? (See https://pubs.acs.org/doi/full/10.1021/acs.jcim.4c01125).

- Why include only arbitrary parts of the mini-G protein? This apparently resulted in such immense artifacts and unrealistic G protein-coupled complexes that the atoms had to be kept in place via positional restraints during the simulations. How can these results be trusted? Indeed, the experimental mini-Gαq proteins in PDBs 6WHA and 7RAN are Gq/Gi/Gs-chimeras, but these mutations could be reverted with a template-based homology modeling. Also, there are templates available with the natural (non-chimera) mini-Gαq-subunit and even with the complete Gαq subunit. However, if the authors had chosen to build an AlphaFold Multimer model of the Gq-coupled 5-HT2A, (see https://www.biorxiv.org/content/10.1101/2021.10.04.463034v2, available and open source since 2022), it would be less artificial than the method applied in this manuscript. Has this G protein-restrained construct been validated in comparison to the non-truncated mini-G protein?

- There is no analysis of MD simulation convergence. The RMSD of the receptor (all residues and residues in the 7-TM domain), of this G protein “construct”, and especially the RMSD of ligands should be shown. Are the ligands still bound to the receptor at the end of the simulations?

- The MD simulations are inconsistent. Why do the simulations have different lengths? This is not explained in the manuscript. How are frames selected/sampled for the analysis? Hopefully, the same number of frames and the same frame selection criteria are applied for all simulation systems (despite different simulation lengths), or else the analyses would be biased to the systems with more frames.

- All simulations should have the same length and be performed in triplicate, using different random seeds for the initial velocities, to increase the statistical significance of observations and to assess the robustness and reproducibility of the findings. Use the RMSD plots to (consistently) remove from your analysis the initial frames in which the systems are not equilibrated.

- Why use 5 different PDB templates that introduce uncertainty and variability to the starting models (due to their different sequences, crystallography artifacts, etc.), while they actually represent only two receptor states – the active/G protein-coupled state and the inactive – just look at their RMSD to one another. Instead, it would be more valuable to focus on just two properly prepared models for the active and inactive states, and then cross-dock and simulate the distinct ligands using these two starting references. In the current manuscript, it is hard to discriminate the effects of the ligands from the effects of the various receptor templates. For example, PDB 7WC9 has a partial agonist bound, but the receptor is in the inactive state due to the Bril-fusion. What does this receptor template mean in the activation energy surface? Probably nothing, it is a crystallography artifact, so why include this?

- PCA is a powerful tool for dimensionality reduction and, via the loading plots, a useful tool to identify the most significant factors contributing to the observed variance. However, just describing the PCA and loading plots does not add to the scientific knowledge about the 5-HT2A receptor. Instead, the authors should use the plots to identify the most significant features and then assess how these features evolve throughout the different simulations – i.e., show the values trajectory and compare them between model systems, applying a statistical test (e.g., ANOVA) to evaluate if the observed differences are statistically significant.

- There is no description or analysis of ligand-protein interaction and how those could impact or be linked to the observed PCA outcomes, or how they vary among the distinct ligand modalities. Perhaps the ligand-receptor interaction could be included among the PCA variables.

- The manuscript is confusing and poorly written (please, check spelling and grammar). Imprecise and unscientific language is frequently used. For example: “left out” L142, “by means of trial and error” L154, “Interpretations… were performed in VMD” L194, “after some equilibration” L435, “Therefore, the question becomes more academic” L488.

- The eigenvalues of the correlation matrices are not shown.

- P220: It is not correct to conclude, based on the PCA of distances, that the residue motions are correlated. The correct approach is dynamic cross-correlation or covariance analysis over the trajectories, not an arbitrary output metric.

- L213: What is the relevance of having the PCA performed on all the trajectories versus the PCA of the coupled and non-coupled receptor? This is not discussed further in the manuscript, and, ultimately, the same data is being analyzed twice.

- L221-223: There is no reference to TM5 in any of the graphs in Figure 2A-C, and nothing in these figures supports the conclusion that “relevant active and inactive states were sampled”. How is this being assessed?

- L242: It is not possible to assess or conclude via these computational experiments that the "differential nature of the ligands has a direct impact on these degrees of freedom" as these observed differences could simply be a result of the different starting points (distinct initial receptor conformations obtained from the experimental structures and their artifacts) rather than a change promoted by the ligands – as the authors admit in L263 – indicating the complexes are “stuck” around local minima. The study lacks control simulations to isolate the effects of specific ligands or the presence/absence of G proteins, making it harder to directly attribute observed differences to specific factors.

- L280: The plots for these systems in the simulations with the G protein-coupled complexes should also be shown and then compared to the “decoupled” systems.

- L368: The bulge in TM5 is an intrinsic feature of the 5-HT2A receptor encoded in the sequence (look at the sequence alignment of the 5-HT receptor family), which possesses a π-helix caused by the extra residue S242 – therefore, it is visible in all experimental structures, independent of the bound ligand or receptor state. It is worrisome to read that the bulge “disappeared” in your simulation (L371). This suggests a poorly prepared system. Please, reevaluate your computational experiments.

- Figure 5 shows that the density profiles of “zotepine + G” and “25CN-NBOH + G” are very similar despite the opposite pharmacological profile of these compounds. From this, the authors conclude, without experimental support, that zotepine locks the receptor in both the active and inactive conformation, i.e., hinders the conformational freedom of the receptor regardless of the initial state. However, one could argue instead that the shift towards the inactive state observed for the other G protein-coupled system (Apo, IHCH-7086, (R)-69) might reflect the unbinding of the G protein construct or artifacts caused by this unnatural construct – this should be properly assess before any conclusion could be drawn from these simulations.

- Having identified this pair of distances (TM3-7, TM2-6) as key descriptors (and predictors) for the modality and pharmacological profiles of ligands, this should be validated by i) statistical analysis, to ensure these differences are indeed significant; ii) simulating ligands that were not included in the previous analysis (i.e., a set of test ligands) to check if their density profiles would match the expected/predicted profile disclosed in Figures 5 and 6. Some obvious choices would be serotonin (by definition, a full and unbiased agonist), LSD (β-arrestin biased, partial agonist and psychedelic, see: https://doi.org/10.1016/j.cell.2020.08.024), Lisuride (β-arrestin-biased, partial agonist, non-psychedelic, see: https://doi.org/10.3389/fmolb.2023.1233743), Pimavanserin (inverse agonist, see: https://doi.org/10.1021/acs.jmedchem.4c01244), all of which been experimental coordinates available. Other alternative validation compounds could be those published at https://doi.org/10.1038/s41467-025-57956-7

- L455-470: None of the presented data supports the claims in this paragraph. There is no evidence that 25CN-NBOH is a psychedelic in humans; it actually produces fewer head-twitch responses (HTR) in mice than other psychedelics (https://doi.org/10.1007/s00213-014-3739-3). Serotonin is not a weak binder (Kd = 1.3 nM, see https://doi.org/10.1016/0006-2952(95)02122-1).

- L472-495: The presented data and results do not support the conclusions. There is no indication that any of the studied compounds have either hallucinogenic or antidepressant effects in humans, as they have not been in a clinical trial. Associating the observed differences in the probability density plots of a couple of CVs (which have not been assessed for convergence and adequate sampling of the phase space) with behavioral effects is overstretching and overinterpretation of the computational results. The alleged equivalence of a small (“microdosing”) of a full agonist to the effect of a partial agonist ignores ligand residence time, receptor turnover, receptor desensitization, and signal amplification mechanisms, while also being unrelated to the results discussed in the manuscript.

Minor comments

- In recent years, the term “psychedelics” has been preferred instead of “hallucinogens”. See https://doi.org/10.1055/a-1310-3990

- In the introduction, he research question and hypothesis should be made explicit. Currently, the aim of the study does not become fully clear until later in the methods and discussion sections.

- How much of the variance can be attributed to each PC? If PC1 and PC2 capture most of the variance, there is no reason to show or mention other PCs, as these might not be relevant.

- L242: Incorrect figure reference.

- Splitting the manuscript into subsections is unnecessary (or relevant). For example, splitting the introduction into 1.1 and 1.2 sections.

- P2L47: Those abbreviations need to be spelled out – not obvious to all readers.

- L206: What is the meaning of “plethora of microscopic modulations”? This does not make sense.

- Since Figure S5 is highly discussed, it should be somehow included in the main document.

- Table 1 should be better organized. For example, grouped into “coupled” and “non-coupled” complexes, color-coded by the state of the receptor in the initial structure (active vs inactive), with ligands sorted by modality (“pharmacological effect”). A structure with a “None” ligand is called “apo”. Spelling and capitalization mistakes should be addressed.

- In all figures displaying structures, hide the non-polar hydrogens. Increase the contrast between cartoon and residue lines (e.g., color them differently) or consider rendering with another tool like ChimeraX or Pymol rather than VMD.

- In all figures, add labels to more quickly discriminate between coupled and non-coupled receptor states.

- Figure 1: Panel A is not relevant (common knowledge on GPCR activation). Panel B should be enlarged, colors adjusted to increase the contrast of the highlighted key residue sets versus the receptor; and hide nonpolar hydrogens. Instead of showing the microswitches, display the most relevant metrics/distances that are discussed in the manuscript (focused on the features with the most significant loadings from Figure S2). Those are what readers should understand and remember. Improving this figure to show the most relevant DOFs (and only those) can replace Table 1, which is already repeated with further detail in Table S2. The refer

- Figure 2: Add labels on top of the graphs to more clearly identify the PCA of all systems, coupled and non-coupled receptors. The caption is confusing. Show first the PCA plots and then the loading plots (with arrows).

- Figure 4: Reorganize the panel to make better use of the page space and increase the figure sizes. Since the χ angles of tryptophan and phenylalanine are discussed heavily across the manuscript, display these in a schematic as they might not be obvious to all readers.

Reviewer #3: This study conducted long-term molecular dynamics simulations of ligand-complexes of the 5-HT2A receptor with antagonists, inverse agonists, G protein-biased agonists, and β-arrestin-biased agonists. The authors analyzed the motion differences in the transmembrane regions and related motifs when the 5-HT2A receptor binds to different types of ligands, contributing to the understanding of the biased activation and antagonism mechanisms of the 5-HT2A receptor. The results should be of help to drug discovery efforts targeting the 5-HT2A receptor.

Suggestions for improving the manuscript are:

1. Further exploration of the interaction differences between related ligands and protein amino acids would be more helpful in elucidating the mechanisms by which different types of ligands influence protein conformation.

2. The article addressed the disconnected regions of the protein (e.g., TM5-6) by capping (without completing the disconnected regions) but did not discuss whether the missing loop segments would affect the simulation results.

3. P3, line 67, the authors state that “(R)-69, which is biased to G protein pathway activation (Emax=87% relative to 5-HT) without any β-arrestin2 recruitment”. This is incorrect. Although compound (R)-69 has been described as G protein-biased, it showed potent activation of the β-arrestin signaling.

4. Figure 1, compounds IHCH-7086 and (R)-69 have designated chirality, such information should be presented.

5. Figure 3: The color scheme lacks sufficient contrast. Consider using both color and shape coding for clearer distinction.

6. The names of amino acid residues are inconsistently represented, with a mix of three-letter and single-letter codes used.

**Have the authors made all data and (if applicable) computational code underlying the findings in their manuscript fully available?**

Reviewer #1: Yes

Reviewer #2: None

Reviewer #3: Yes

PLOS authors have the option to publish the peer review history of their article (what does this mean? ). If published, this will include your full peer review and any attached files.

**Do you want your identity to be public for this peer review?** For information about this choice, including consent withdrawal, please see our Privacy Policy .

Reviewer #1: **Yes: ** Brunoo Guiard

Reviewer #2: No

Reviewer #3: No

**Figure resubmission:**

**Reproducibility:**



---

## [Decision Letter · Decision Letter 1]

28 Jul 2025

PCOMPBIOL-D-25-00549R1

Molecular dynamics study of differential effects of serotonin-2A-receptor (5-HT2AR) modulators

PLOS Computational Biology

Dear Dr. Vanommeslaeghe,

Thank you for submitting your manuscript to PLOS Computational Biology. After careful consideration, we feel that it has merit but does not fully meet PLOS Computational Biology's publication criteria as it currently stands. Therefore, we invite you to submit a revised version of the manuscript that addresses the points raised during the review process.

Please submit your revised manuscript within 30 days Sep 27 2025 11:59PM. If you will need more time than this to complete your revisions, please reply to this message or contact the journal office at ploscompbiol@plos.org. Please include the following items when submitting your revised manuscript:

We look forward to receiving your revised manuscript.

Kind regards,

David van der Spoel

Academic Editor

PLOS Computational Biology

Nir Ben-Tal

Section Editor

PLOS Computational Biology

**Reviewers' comments:**

Reviewer's Responses to Questions

Reviewer #2: The authors have addressed my comments and concerns satisfactorily, and the manuscript can be recommended for publication. I appreciate the effort they put into implementing my critique and suggestions, and the thoroughness with which they answered my questions. Although there were some points of disagreement, which is entirely expected in the scientific process, the revisions reflect meaningful engagement with the review. The manuscript has significantly improved as a result, making a valuable contribution to our understanding of the 5-HT2A receptor.

Reviewer #3: The authors have adequately addressed all my concerns in the revised manuscript, and I recommend its acceptance for publication.

However, prior to final publication, I suggest including a brief discussion of a very recent study (https://doi.org/10.1016/j.neuron.2025.06.012) in the Introduction section. This work challenges the notion of direct psychedelic-TrkB binding, demonstrating no observable TrkB activation across a comprehensive panel of 41 psychedelics. Given the significance of this finding to the broader field, a mention of this work—even as a point of ongoing debate—would be necessary.

**Have the authors made all data and (if applicable) computational code underlying the findings in their manuscript fully available?**

Reviewer #2: Yes

Reviewer #3: Yes

PLOS authors have the option to publish the peer review history of their article (what does this mean? ). If published, this will include your full peer review and any attached files.

**Do you want your identity to be public for this peer review?** For information about this choice, including consent withdrawal, please see our Privacy Policy .

Reviewer #2: No

Reviewer #3: No

**Figure resubmission:**
---

## [Editor Report · Decision Letter 2]

21 Aug 2025

Dear Prof. Vanommeslaeghe,

We are pleased to inform you that your manuscript 'Molecular dynamics study of differential effects of serotonin-2A-receptor (5-HT2AR) modulators' has been provisionally accepted for publication in PLOS Computational Biology.

Best regards,

David van der Spoel

Academic Editor

PLOS Computational Biology

Nir Ben-Tal

Section Editor

PLOS Computational Biology

---

## [Editor Report · Acceptance letter]

PCOMPBIOL-D-25-00549R2

Molecular dynamics study of differential effects of serotonin-2A-receptor (5-HT2AR) modulators

Dear Dr Vanommeslaeghe,

I am pleased to inform you that your manuscript has been formally accepted for publication in PLOS Computational Biology. Your manuscript is now with our production department and you will be notified of the publication date in due course.

With kind regards,

Judit Kozma
